# Digitally-enhanced lubricant evaluation scheme for hot stamping applications

Xiao Yang [1,2], Heli Liu[1,2], Saksham Dhawan[1,2], Denis J. Politis[3], Jie Zhang [1], Daniele Dini [1], Lan Hu[1,2], Mohammad M. Gharbi[4] & Liliang Wang [1,2] ✉

Digitally-enhanced technologies are set to transform every aspect of manufacturing. Networks of sensors that compute at the edge (streamlining information flow from devices and providing real-time local data analysis), and emerging Cloud Finite Element Analysis technologies yield data at unprecedented scales, both in terms of volume and precision, providing information on complex processes and systems that had previously been impractical. Cloud Finite Element Analysis technologies enable proactive data collection in a supply chain of, for example the metal forming industry, throughout the life cycle of a product or process, which presents revolutionary opportunities for the development and evaluation of digitally-enhanced lubricants, which requires a coherent research agenda involving the merging of tribological knowledge, manufacturing and data science. In the present study, data obtained from a vast number of experimentally verified finite element simulation results is used for a metal forming process to develop a digitally-enhanced lubricant evaluation approach, by precisely representing the tribological boundary conditions at the workpiece/tooling interface, i.e., complex loading conditions of contact pressures, sliding speeds and temperatures. The presented approach combines the implementation of digital characteristics of the target forming process, data-guided lubricant testing and mechanism-based accurate theoretical modelling, enabling the development of data-centric lubricant limit diagrams and intuitive and quantitative evaluation of the lubricant performance.

The digital transformation and Industry 4.0 technologies are rapidly shaping the future of manufacturing. The key to success for digital manufacturing is the holistic approach in connecting all essential technologies across an organisation with strategic partners instead of following traditional isolated implementations. Digitally enhanced manufacturing is closely associated with various emerging technologies and approaches, such as cyber-physical systems (CPS), the Internet of Things (IoT), big data and cloud computing, that have driven conventional manufacturing towards a strategic revolution based on the collaboration of information and technology, leading to the enhancement of efficiency, sustainability, quality control, and flexibility. By 2025, it is estimated that the global value of IoT technologies will reach \$6.2 trillion[1]. In general, CPS refers to a system that establishes real-time data exchange between the physical entity and its virtual replication (usually known as digital twin)[2,3]. This system can acquire, restore, analyse, and feedback data based on IoT, which can realise collection of historical data (leading to big data), immediate fault localisation, supervision of machinery health status and

[1]Department of Mechanical Engineering, Imperial College London, London SW7 2AZ, UK. [2]SmartForming Research Base, Imperial College London, London SW7 2AZ, UK. [3]Department of Mechanical and Manufacturing Engineering, University of Cyprus, 1678 Nicosia, Cyprus. [4]Houghton Deutschland GmbH, Giselherstraße 57, 44319 Dortmund, Germany. ✉e-mail: Liliang.Wang@imperial.ac.uk

maintenance requirements. These digital technologies have triggered revolutionary changes across the manufacturing industry[4–6] and integration of these techniques can significantly promote the digital transformation of a modern manufacturing facility to enhance innovation and competitiveness. As illustrated in recent research, by leveraging cloud finite element analysis (FEA) for real-time connectivity and advanced analytics to create a connected ecosystem, a low cost IoT solution can be offered to manufacturing industries with access to proactive and reliable market demand data. These data provide an exciting opportunity to produce digitally enhanced products[7,8].

One sector that has the potential to benefit greatly from these emerging technologies is the metal forming industry. Over 160,000 engineering materials and nearly 90%[9] (wt. %) of products made from metals, are produced by metal forming technologies, thus resulting in metal forming being one of most widely used manufacturing crafts. Steel and aluminium, being some of the most commonly utilised metals, have seen demand rise to over 1.4 billion and 45 million tonnes per annum, respectively, with approximately 90%[10] and 67%[11] of steel and aluminium products being manufactured by at least one metal forming processes. During the last two decades, computer-aided finite element (FE) simulations have been introduced into the metal forming sector to achieve process optimisation and reduce the cost of trial-and-error prototyping. Suitable material characterisation and accurate boundary condition definitions are essential prerequisites for a robust FE prediction of material behaviour and sophisticated forming conditions[12]. Considerable efforts have been made by researchers to expand the functionalities of FE analysis to address the different problems encountered in metal forming, such as insufficient formability, poor surface quality, inadequate quenching and post-form strength as well as springback[13–19]. The wide utilisation of FE models in various manufacturing sectors makes it a promising and appropriate source of data to accelerate the transformation to digital manufacturing and thus increase the efficiency of the manufacturing system, including the metal forming industry.

During metal forming processes, where galling and abrasive wear occur on the tooling, lubricant is usually applied at the tool–workpiece interface to separate and protect the contacting surfaces, reduce forming loads and increase the tooling life. Contrasting to the situation where the tribological system is lubricated by the oil bath or supplied with the lubricant at a constant rate, which usually occurs in mechanical systems such as transmissions[20,21], engines and bearings[22,23], lubricant is applied before the metal forming process commences and cannot be replenished during the process. This may lead to possible lubricant breakdown and thus surface damage to the tooling and even failure in the formed component if insufficient lubricity is provided. Therefore, the appropriate selection of lubricant is essential for the successful production of qualified components at optimal cost.

There are various types of lubricants available ranging from straight oils and water-based lubricants to liquid-solid blenders and solid lubricant coatings, which can meet a wide range of requirements depending on the application. However, the selection of lubricant applied in a specific forming process is usually experience-oriented and based on the evaluation results obtained by the lab-scale friction tests. Although the lab-scale friction test has the advantages of easy accessibility due to simplified geometry of tooling and specimen and decreased equipment capacity, it is difficult to reproduce the tribological contact conditions experienced in the actual forming process, which reduces the robustness and reliability of the testing results. In addition, the conventional friction characterisation technique usually yields a constant friction value to represent a complex tribological phenomena on the tool–workpiece interface, which is not suitable and will cause inaccuracy in the predicted results.

In this work, data obtained from a vast number of experimentally verified finite element simulations for a metal forming process, e.g. hot stamping, are used to develop a data-centric and digitally enhanced lubricant evaluation approach, by precisely representing the tribological boundary conditions at the workpiece/tooling interfaces, to enable an informed decision to be made on the lubricant used for the target forming process. Specifically, such variables incorporate rapid changes and complex loading of contact pressures, sliding speeds and temperatures. This approach combines the implementation of advanced theoretical modelling tools and cutting-edge experimental methodologies with data-centric approaches for the optimisation of manufacturing processes, which will enable the next generation of digitally enhanced design tools, leading to more accurate process optimisation and tailor-made products to be developed with potential for deployment across multiple manufacturing industries.

## Data review and digital characteristics (DC) of hot stamping

DC is proposed for characterising manufacturing technologies from the perspective of the metadata generated during the process. The DC is the visualisation of manufacturing metadata for a specific manufacturing process incorporating essential information spanning the design, manufacturing and application stages of manufactured products. DC of a specific metal-forming process is a crucial and comprehensive collection of data originating from practical experimentation and associated experimentally verified FE simulations, which includes distributions and evolutions of process parameters and contact conditions, such as stress, strain, strain rate, contact pressure, relative sliding speed and interfacial temperature for every element and every stage of the manufacturing process.

The DC of the hot stamping process is developed based on the metadata, which is provided by a cloud-based repository for knowledge transfer and data sharing, which contains open access data sets for research purposes[24–26]. This metadata set containing multiple simulations has been contributed by developers whose work has been previously reported in peer-reviewed studies over the past 15 years[17,27–30]. The entire metadata set covers the manufacturing processes of a wide range of hot-stamped components from the FE analysis spanning over 20 types of evolutionary thermo-mechanical parameters including interfacial temperature, contact pressure, sliding speed, effective stress and strain, etc. All of the involved simulations have been experimentally verified to ensure the robustness of the extracted thermo-mechanical parameters and veracity of data.

In terms of tribologically related testing, the DC involves contact pressure, relative sliding speed, interfacial temperature, and relative sliding distance at the tool–workpiece interface. By extracting data from experimentally verified FE simulations and performing subsequent analysis, a comprehensive data visualisation can be generated to demonstrate how contact conditions flow during the forming process and provide guidance for more relevant study.

Tribologically related data on the tool–workpiece interface at each forming step are extracted from the experimentally verified FE simulations. The relative sliding speed experienced by the node on the tool surface accounts for both translational sliding of workpiece material and the surface enlargement resulting from the large deformation. As the sliding time elapsed between the tool and workpiece is dependent on the component geometry and the region where the node is located, a time-based evolution process of contact conditions is transformed into a distance-based version which corresponds to the relative sliding distance at the tool–workpiece interface. The relative sliding distance experienced by the node on the tool surface is obtained by the accumulated material flow through the tool surface in each step and contact pressure, speed and temperature that can be interpolated for each step accordingly. As each forming process of the component has a unique forming stroke, the normalised sliding distance is introduced as the ratio of the absolute relative sliding distance and the forming stroke of the corresponding forming process, to allow processes with different forming strokes comparable. Thus, the

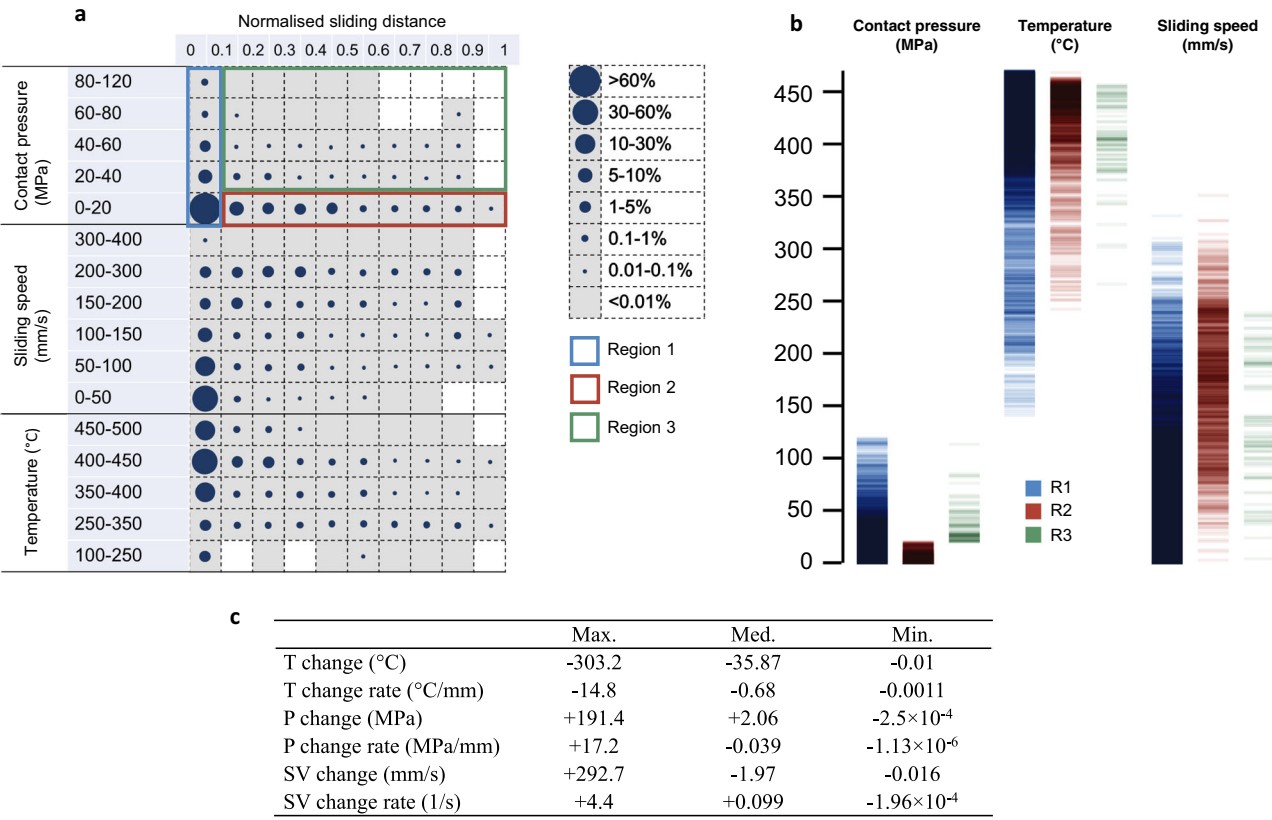

**Fig. 1 | Digital characteristics (DCs) of the hot stamping process. a** Distribution of probability of the tooling elements falling into a specific range defined by the contact condition and normalised sliding distance. Circle sizes indicate the magnitude of the probability. Rectangular frames show the division of different key regions. **b** Distribution of contact condition values in different key regions, i.e. region 1 (R1), region 2 (R2) and region 3 (R3). **c** Maximum (Max.), median (Med.) and minimum (Min.) values of change and change rate of contact condition, i.e. temperature (*T*), contact pressure (*P*) and sliding speed (*SV*), during hot stamping evaluated and listed.

|  | Max. | Med. | Min. |
|---|---|---|---|
| T change (°C) | -303.2 | -35.87 | -0.01 |
| T change rate (°C/mm) | -14.8 | -0.68 | -0.0011 |
| P change (MPa) | +191.4 | +2.06 | $-2.5×10^{-4}$ |
| P change rate (MPa/mm) | +17.2 | -0.039 | $-1.13×10^{-6}$ |
| SV change (mm/s) | +292.7 | -1.97 | -0.016 |
| SV change rate (1/s) | +4.4 | +0.099 | $-1.96×10^{-4}$ |

evolution of contact conditions as a function of the normalised sliding distance are mapped to describe the tribological characteristics of a specific metal forming process.

Analysis of tribologically related DC at the tool–workpiece interface in the hot stamping process was conducted by plotting the probability of elements falling into the specific ranges determined by contact pressure, relative sliding speed, interfacial temperature, and normalised sliding distance, as shown in Fig. 1a. For the hot aluminium stamping process, over 80.2% of the elements experience the contact pressure below 20 MPa. Approximately 12.2% of elements fall into the range of 20 – 140 MPa with the normalised sliding distance lower than 0.1. Over 54.5% of elements are found to be below 50 mm/s with the normalised sliding distance up to 0.7. High probability of 15.5% are also observed in the range of 150–300 mm/s. Although 57.8% of elements are concentrated in the interfacial temperature over 400 °C, approximately 4.4% experiences a relatively low temperature range of 100 to 250 °C.

The elements can be divided into three key regions according to the unique tribological features, i.e. short-stroke region (region 1), long-stroke region (region 2) and hybrid region (region 3), as shown in Fig. 1a. The short-stroke region (R1) features with relatively short sliding distance (<0.1) and a large range of contact pressure (up to 200 MPa). In contrast, the long-stroke region (R2) demonstrates a large normalised sliding distance (up to 1) with low contact pressure (lower than 20 MPa). The hybrid region (R3) is also known as the tribologically harsh region since both contact pressure and sliding distance values were high. Significant deviations of contact pressure, sliding speed and interfacial temperature can be observed in different key regions, as demonstrated in Fig. 1b, which would affect the lubricant behaviours and friction evolutions to a large extent.

In addition, it has been found that complex loading of contact conditions widely exists during the hot stamping process. The probability for experiencing changes in the contact conditions at mating interfaces is over 99% for complex loading conditions and the evolution of pressure, speed and temperature is widely observed[30–33]. The maximum, median and minimum change and rate of change for contact condition, i.e. interfacial temperature (*T*), contact pressure (*P*) and relative sliding speed (*SV*) can be observed in Fig. 1c. By taking this complex loading feature of contact conditions into account, a data-centric testing approach has been generated to guide the subsequent friction tests. Here 'data-centric' approach indicates a significant transformation from traditional 'experience-oriented' approaches for how research is conducted and information is processed. It encompasses extensive data collection and state-of-the-art data storage techniques to extract insightful data analytics and comprehensively accelerate the implementation of ideas into applications. This provides an unprecedented opportunity in complementing traditional experience-oriented research approaches conventionally used in fields such as manufacturing with concepts and emerging methods deduced from data[34].

Following this data-centric approach, preliminary research revealed that the complex loading conditions, featuring rapid temperature, pressure and speed changes in mating interfaces, significantly influenced the transient lubricant behaviour and led to remarkable changes in coefficient of friction (COF) values and also to premature lubricant breakdown[35,36]. An accurate, mechanism-based interactive friction model was subsequently developed to predict how lubricant and surfaces respond to the transient contact conditions present in the forming process, which can describe and evaluate the

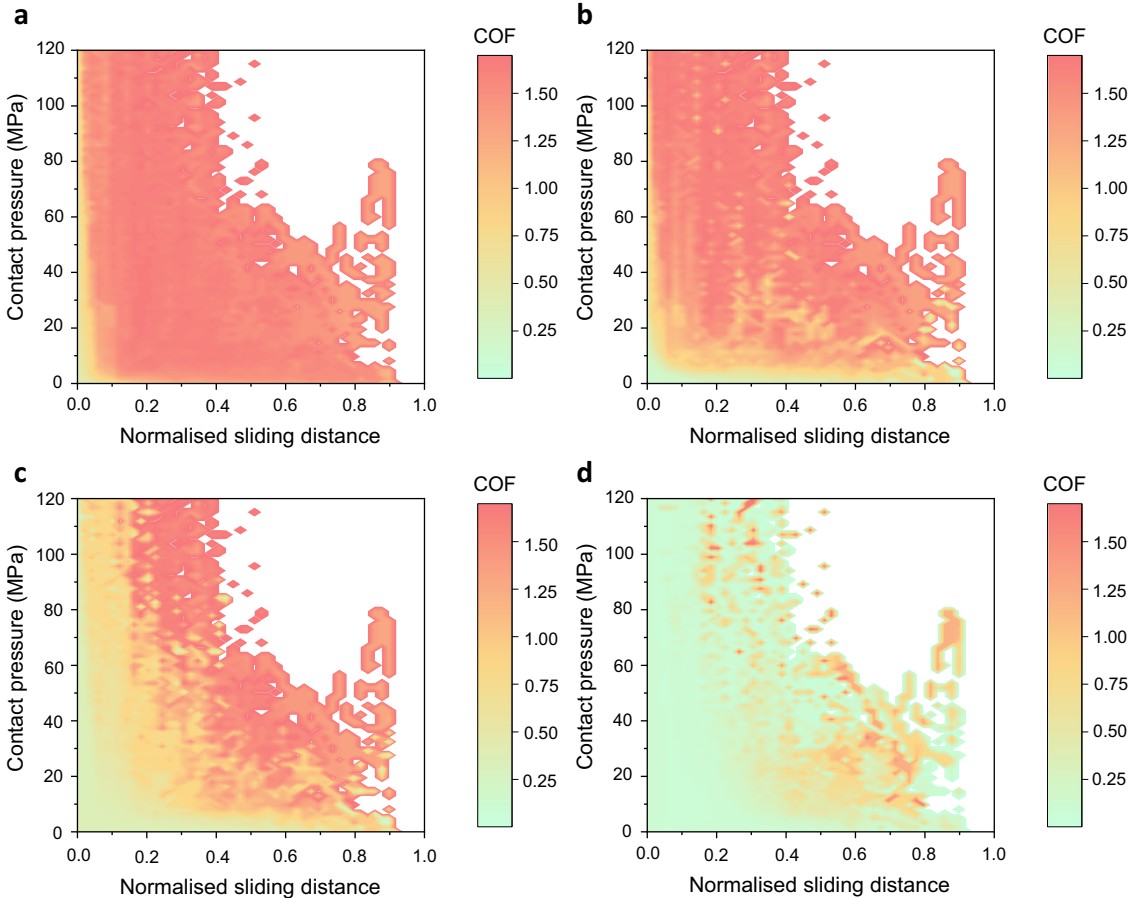

**Fig. 2 | Digitally enhanced evaluation of four lubricants applied in the hot stamping process via the lubricant limit diagram (LLD) demonstrated by the** coefficient of friction (COF) values. **a** Lubricant #1. **b** Lubricant #2. **c** Lubricant #3. **d** Lubricant #4.

lubricant performance in a reliable and efficient manner (see Methods).

### Digitally enhanced lubricant evaluation for hot stamping

The data-centric lubricant limit diagram (LLD) is proposed to map the lubricant behaviours following the DC of an application scenario based on the mechanism-based theoretical interactive model (see Methods). In this research, the performance of four lubricants developed for the hot aluminium stamping process were evaluated following the data-centric approach and comprehensively demonstrated by LLDs, which is the digitally enhanced lubricant evaluation process. As shown in Fig. 2, the LLDs of four lubricants were demonstrated by the COF values as functions of the contact pressure and normalised sliding distance. It can be observed that as contact conditions become severe, i.e. contact pressure and sliding distance increased, the lubricant loses its efficacy gradually and COF increases as lubricant breakdown occurs. The overall lubricant performance is negatively correlated to the area ratio of the red region to the whole. Thus, lubricant #4 has the most desirable performance among these four lubricant candidates. Although lubricant #3 presents larger tribologically safe area (approximately 70%), which means the lubricant has sufficient lubricity to protect the contact surfaces, compared to lubricant #2 (approximately 37%), the COF value before the lubricant breakdown occurs is higher (around 0.3, indicated by the lighter green colour in the low contact pressure region, e.g. <10 MPa).

Instantaneous COF values are not only related to contact condition evolutions, but dependent on interfacial characteristics, including lubricant properties and surface roughness of contacting counterparts.

To establish a comparable standard and enable a simple performance evaluation between different lubricants, an LLD is developed to demonstrate the lubricant performance grade (*G*) which represents a quantified evaluation parameter of the lubricant performance under the applied contact conditions. This value considers the increase of COF value at the interface as a criteria (see detailed calculation in Eq. (1), 'Methods'). It should be noted that both COF values and the performance grade are calculated based on the relative sliding distance (in 'mm') instead of the normalised sliding distance.

As shown in Fig. 3, the LLDs of four lubricants are demonstrated by the performance grades as functions of the contact pressure and normalised sliding distance. An overall performance grade can also be calculated (OPG, being the average value calculated by the performance grade of each contact condition) for each lubricant applied to the target forming process. In this study, lubricant #1 gives a low OPG of 17.5% while lubricant #4 has a much more desirable performance of close to 100%, which means that this lubricant is ideally developed for the target hot aluminium stamping process. Lubricant #2 and #3 have medium OPG of approximately 35 and 65%, respectively. By intuitive mapping and quantification of the lubricant performance, the LLDs can provide reliable evidence for lubricant selection of a target application scenario.

There are five regions that are differentiated based on the tooling geometry and forming features in the hot stamping process, namely, the blank holder region, die corner region, flat bottom region, side wall region and die shoulder region, as shown in the schematic diagram of Fig. 4e. It has been found that, for the die corner region, approximately 50% of elements experience a high interfacial temperature of over

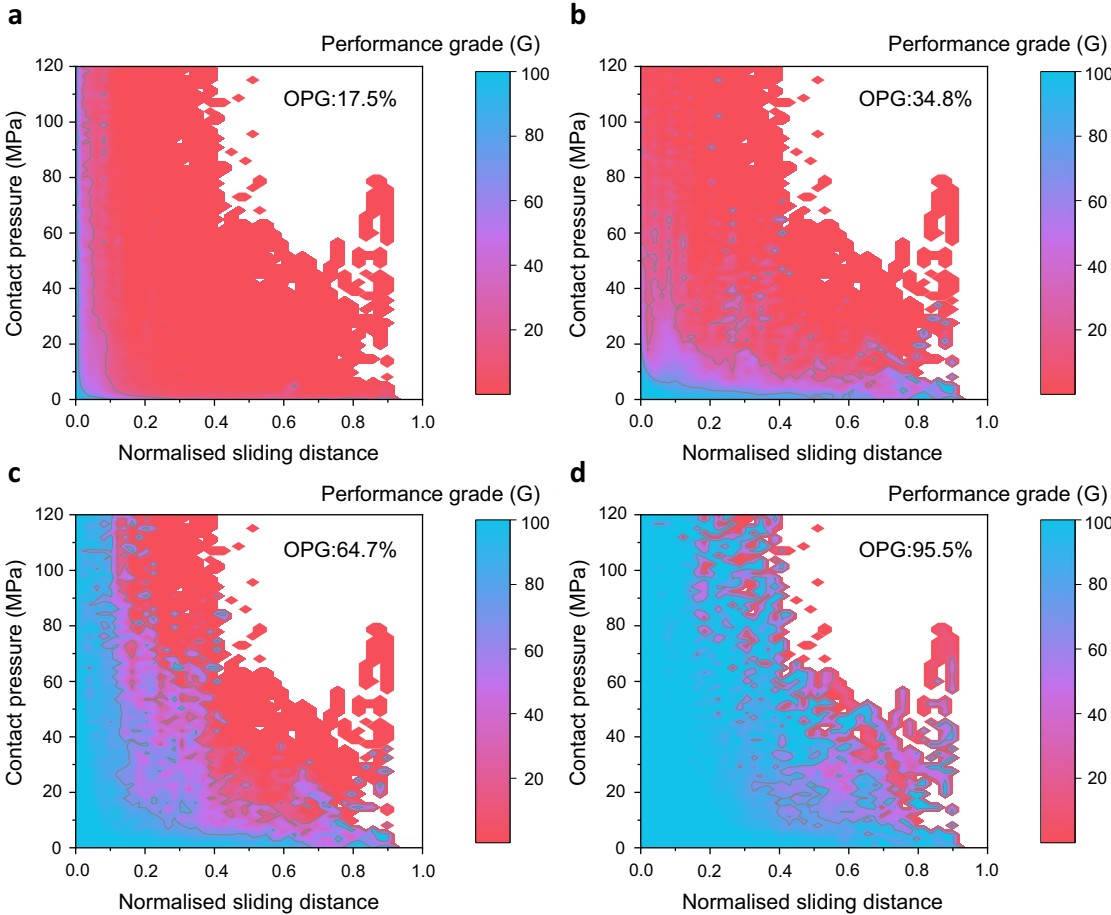

**Fig. 3 | Digitally enhanced evaluation of four lubricants applied in the hot stamping process via the lubricant limit diagram (LLD) demonstrated by the** **performance grade (G) and comparison by the overall performance grade (OPG). a** Lubricant #1. **b** Lubricant #2. **c** Lubricant #3. **d** Lubricant #4.

450 °C and elements experiencing contact pressure greater than 10 MPa account for 34%. For the flat bottom region, pressures lower than 5 MPa represent the majority of elements (70%) while the normalised sliding distance of over 0.1 is found in only 3.2% which means that most elements experience very short sliding distance (Contact condition distribution for the individual forming region shown in Supplementary Fig. 8). In addition, the original percentage of elements in the die corner region is approximately 27%, which is much greater than 11% of the flat bottom region. This means that more elements in the die corner would possibly fail if under the same contact conditions. All these factors may finally lead to the results where the contribution of the die corner region to lubricant failure is greater than the flat bottom region.

Data distribution of the lubricant failure region, which is indicated by red colour in LLDs, has been investigated. As shown in Fig. 4, the ring charts show the data distribution results of elements with grades less than 30% for each lubricant. The number in the centre denotes the area ratio of the red region to the whole region of the target forming process. Thus, the decrease of this number indicates an increase in the corresponding lubricant performance.

It can be observed that although the detailed numbers are different, the die corner region always contributes the most to the lubricant failure area, from 34 to 19%, and the flat bottom region the least. In some circumstances it may not be the least (as for lubricant #2), but the proportion is much smaller compared to the die corner region.

The effect of initial lubricant volume on the lubricant performance is also investigated. The OPGs of each lubricant under a series of initial

lubricant volumes from 5 to 100 g/m² are presented in Fig. 5. In general, it has been found that the overall performance grade increases as the initial volume increases for the three lubricants (#1–#3). For lubricant #3, it can be observed that a large increase of overall grade occurs from approximately 40 to 70% when the initial lubricant volume increases from 5 to 100 g/m². However, the increase becomes negligible when a certain value is reached. For lubricant #3, this value is approximately 80 g/m² and the optimised grade is 68%. A similar phenomenon can be found for both lubricants #1 and #2, although for lubricant #2, the maximum value is achieved much earlier at approximately 40 g/m² with the maximum grade of approximately 35%.

To validate the evaluation results based on the DC and LLD, further experimental testing of the identified most suitable lubricant was conducted by using an industry scale production system to perform hot stamping of an automotive component (Supplementary Fig. 4). An environmentally friendly requirement was raised by the end user. Thus, lubricant #4 was not considered in this forming test as it is a graphite-based lubricant, which does not meet the environmental specifications. The most suitable lubricant candidate was selected from lubricants #1–3, which are water-based lubricants. Based on DC and LLD analyses, the OPG of lubricants #1, #2 and #3 for this 'side beam' forming test were 78.4%, 75.8% and 97.3%. Data distribution of the lubricant failure region was also investigated for each lubricant (Supplementary Fig. 5). Therefore, lubricant #3 was the best candidate to be applied during this forming process. When lubricant #1 or #2 was applied, moderate scratches were observed on the side wall/blank holding area. While after the application of lubricant #3, scratches

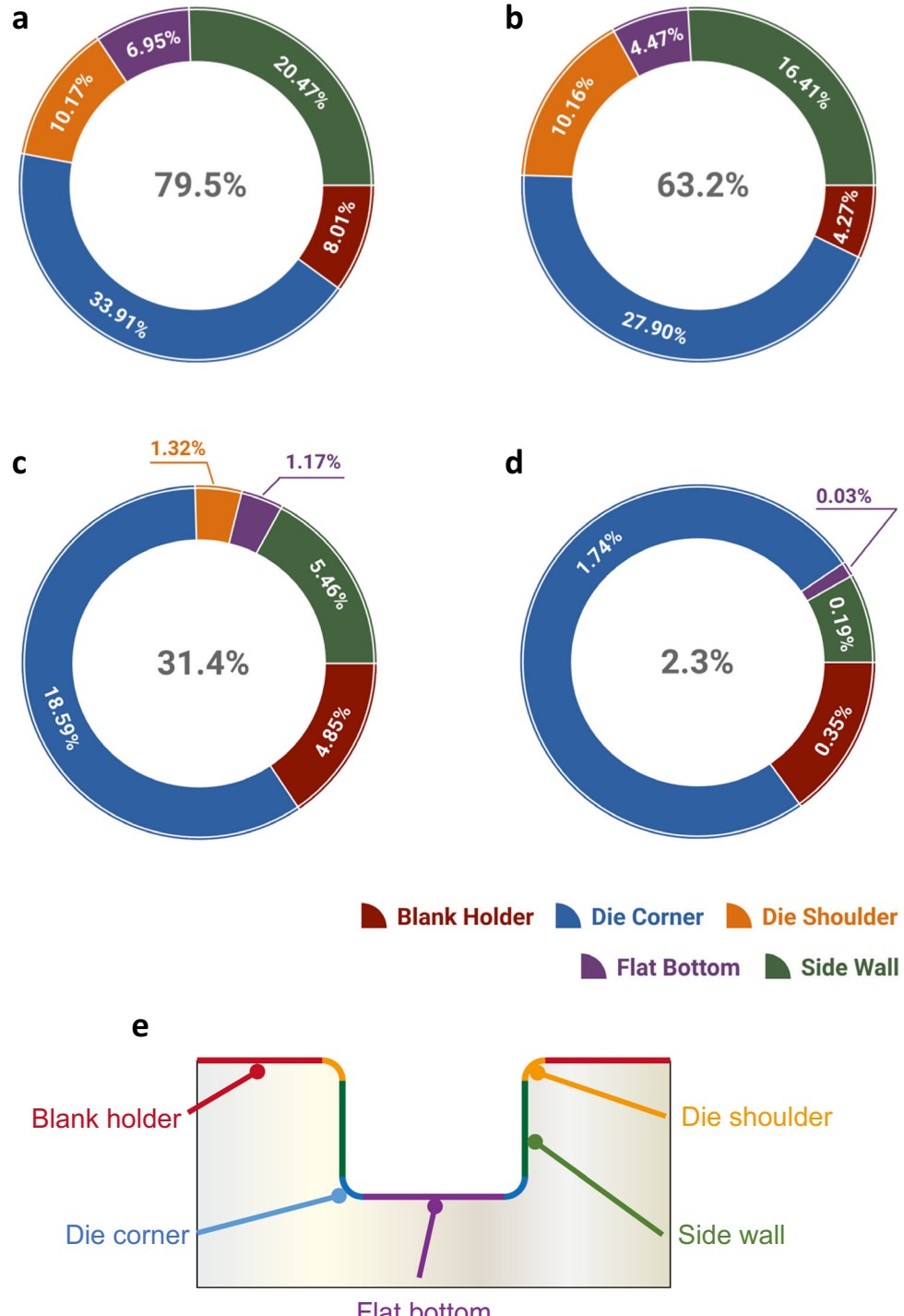

**Fig. 4 | Digitally enhanced evaluation of four lubricants applied in the hot stamping process via data distribution analysis of the lubricant failure area (grades less than 30%). a** Lubricant #1. **b** Lubricant #2. **c** Lubricant #3. **d** Lubricant #4. **e** Schematic diagram of five different regions based on the tooling geometry and forming features in the hot stamping.

were negligible and surface quality was improved and desirable compared to #1 and #2, which is consistent with the LLD predictions. Validation results are not shown here due to confidential requirements but can be made available from the corresponding author upon request, as stated in the 'Data availability' section.

## Discussion

In recent years, smart manufacturing and associated digital process transformation has demonstrated a significant potential in accelerating production and technology developments. This has also recently transitioned into the lubricant development field. Inspired by this trend towards smart manufacturing, the present paper proposes a digitally enhanced lubricant evaluation scheme based on the DC of a target manufacturing process, i.e. hot stamping, utilising the data-centric LLD. The DC utilised big data to characterise the target manufacturing process, incorporating essential information spanning all stages of production. Specifically, it presented an integral overview of the target process variables, such as interfacial temperature, contact

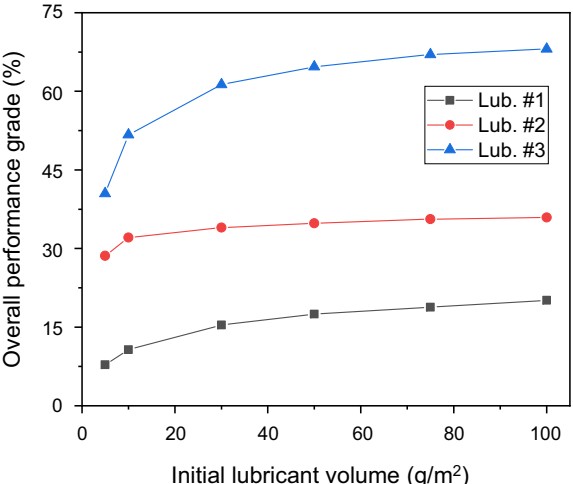

**Fig. 5 | Effect of the initial lubricant volume on the overall performance grade (OPG) of the three lubricant candidates for the hot aluminium stamping process.** Evolution of OPG as the initial lubricant volume increases from 5 to 100 g/m² for the three lubricants, respectively. Detailed values of OPG are shown on the right table.

pressure and sliding speed, from the evolutionary (complex loading) perspective and provided guidance on the subsequent lubricant characterisation and evaluation. Data-centric LLDs were finally developed to enable the accurate simulation and evaluation of lubricant performance throughout the target manufacturing lifecycle based on a predictive, mechanism-based theoretical model, which can also lead to specific identification of dangerous region for lubricant failure based on big data analysis.

The present work has successfully introduced and elaborated the use of data-centric methodologies to create a pipeline for evaluating the performance of widely used lubricants in hot aluminium stamping process, which can further shed light on broader application scenarios and facilitate the achievements made possible by digital manufacturing. However, particular attention should be paid to the data volume (big data) and veracity to ensure a robust and relevant demonstration of DC. It should also be noted that this advanced data-centric methodology necessitates a thorough and deep understanding of how lubricant and surfaces respond to the transient contact conditions present in the target manufacturing process, which is the foundation of a predictive, mechanism-based theoretical model. By linking each model parameter in the predictive friction model with the corresponding physical meanings, improvements of the lubricant performance due to modifications of physical properties and chemical additives can be proactively captured and analysed through LLDs, which will introduce the digitally enhanced lubricant development and accelerate customised product design in the future.

## Methods

### Interactive friction modelling

The mechanism-based interactive friction model has been developed to describe the friction evolution and lubricant behaviours as a function of the instantaneous contact conditions, i.e. contact pressure, sliding speed and interfacial temperature. The capability of this model to accurately predict the lubricant performance under the complex loading conditions, such as rapid changes and complex loading of pressure, speed and temperature, has been thoroughly investigated and validated in the references[35,36].

Friction testing results of four lubricant candidates under constant loading conditions were utilised to calibrate the interactive model parameters and close agreements have been achieved between modelling and experimental results, as shown in Supplementary Fig. 1. A minimum number of friction tests under constant loading conditions

were conducted for the optimisation of both lubricant testing efficiency and model parameter calibration accuracy. According to the conclusions of the references[35,36], the mechanism-based and time-dependent interactive friction model can accurately represent and predict lubricant behaviour under complex loading conditions after the model parameters are calibrated against testing results under constant contact conditions. This advantage of the calibrated model, with time-dependent equations, enables the underlying lubrication mechanism transformation leading to lubricant breakdown to be determined through simplified constant contact conditions. Further information and the corresponding model response under complex loading conditions for the two-phase lubricant are presented in Supplementary Fig. 2 to explain this advanced feature of the interactive friction model.

Interfacial temperature distribution across the sliding wear track was investigated by using Abaqus. The temperature deviation between the contact interface and the measure point is less than 1.3% of the nominated value, which is acceptable and presents minor effects on the final lubricant evaluation results (details see Supplementary Figs. 6 and 7).

Information on the four lubricants is provided in Supplementary Table 1. Calibration of model parameters were conducted by the in situ friction modelling program embedded in Tribo-Mate testing system[35,36]. The optimised parameters of the established interactive friction models for four lubricants are displayed in Supplementary Tables 2–5.

### The data-centric LLD

The data-centric LLD was proposed to map the lubricant behaviour following the DC of an application scenario. The LLD provides a graphical description of the COF evolution following the tribological evolution of tool elements emulating the actual forming procedures based on the interactive friction model. Considering the time consumption and costs involved in performing experimental work, a more efficient and economical method would be to generate the theoretically obtained LLD according to the predictions of the interactive friction model which can consider and reflect the effects of instantaneous contact conditions on the COF evolution results.

A case study of how to apply the developed interactive friction model to the prediction of friction evolution as a function of the instantaneous contact condition, was developed. Supplementary

Fig. 3a demonstrates the evolution history of contact conditions, such as pressure, speed, and temperature, as a function of the sliding distance for an individual element at the tool–workpiece interface in the hot stamping process. As the interactive model is a time-based friction model[35,36], all the contact conditions as the function of sliding distance, namely sliding time, can be input into the model equations to calculate the corresponding COF in each time step, where the evolution of COF can be obtained, as shown in Supplementary Fig. 3b. At the beginning of sliding, the friction value remains at a low level of approximately 0.29 which means the lubricant has adequate lubricity. The slight change of COF is due to the temperature change. After a sliding distance of approximately 24 mm is achieved, COF begins to increase rapidly which indicates that the lubricant breakdown occurs and the lubricant begins to lose its efficacy.

As the friction value at the initial low friction stage is dependent on lubricant properties and the chemical composition, it may be difficult to compare the lubricant performance by directly using friction values. For example, for lubricant #2 the COF value at lubricated condition is <0.1, while for lubricant #3, the value may be as high as 0.28. There is a significant difference in the friction values although both lubricants have good lubricity at this stage. Therefore, it is suitable to use the performance grade instead to evaluate the lubricant performance in a comparable manner. The COF value is transformed to the corresponding lubricant performance grade, based on the expression shown in Eq. (1).

$$G = \begin{cases} 100, \mu(t) = \mu_l(t) \\ 100\,\frac{\mu_d - \mu}{\mu_d - \mu_l}, \mu_l(t) < \mu(t) \le \mu_d(t) \end{cases} \quad (1)$$

where $\mu$ is the instantaneous COF calculated by the interactive friction model, $\mu_l$ is the friction value under lubricated condition and $\mu_d$ is the friction value under dry sliding condition. When the COF value equals to that of the lubricated condition, the performance grade is defined as 100. When the value begins to increase, the grade is determined by the ratio to that of the dry sliding condition. $\mu_d$ and $\mu_l$ are determined based on friction testing results and modelled by using the Arrhenius equation to represent the temperature effects, as described by Eqs. (2) and (3):

$$\mu_d = \mu_{d0}\exp\left(-\frac{Q_d}{RT}\right) \quad (2)$$

$$\mu_l = \mu_{l0}\exp\left(-\frac{Q_l}{RT}\right) \quad (3)$$

where $\mu_{d0}$ and $\mu_{l0}$ are model constants, $Q_d$ and $Q_l$ are activation energy for dry sliding and lubricated conditions, respectively. These parameters were calibrated against the friction testing results under different temperature conditions. As a result, the friction evolution can be transformed to the performance grade evolution as a function of the sliding distance. Before lubricant breakdown occurs, the grade remains at 100 and then decreases rapidly when it loses lubricity, as shown in Supplementary Fig. 3c.

By considering the instantaneous contact conditions experienced by each tooling element based on the DC of the target forming process, the interactive friction model can yield the predictions of lubricant behaviours and calculate the evolution of COF following the exact contact evolution history. By plotting the performance of all the elements in the figure, LLD can provide an intuitive demonstration and quantified evaluation of the lubricant performance for the target forming process.

## Data availability
All the needed information to interpret the findings reported herein can be found within the manuscript and its Supplementary Files. Further experimental validation results and the source data used to generate the Figures and Supplementary Figures are not publicly available for commercial confidentiality reasons but can be made available from the corresponding author upon request, subject to signing an NDA document. A copy of the latter can be obtained from the corresponding author upon request.

## Code availability
The codes generated during this study are available from the corresponding author upon reasonable request.

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

## Acknowledgements

The authors would like to acknowledge the support from the Schuler Pressen GmbH and Novelis.

## Author contributions

L.W. and X.Y. conceived the project. X.Y. performed the experiments and characterisation, data analysis and modelling, wrote the manuscript and addressed the reviewers' comments with significant contributions from D.D. and D.J.P.; H.L. conducted data visualisation and generated figures with help from D.S.; M.M.G., L.H. and J.Z. provided support in conducting experiments.

## Competing interests

The authors declare that they are bound by confidentiality agreements that prevent them from disclosing their competing interests in this work.
