## [Peer Review File · Nature Communications]

Digitally-enhanced lubricant evaluation scheme for hot stamping applicationsEditorial Note: Parts of this Peer Review File have been redacted as indicated to maintain confidentiality.

REVIEWER COMMENTS

Reviewer #1 (Remarks to the Author):

This communication reports a novel approach for digitally enhanced lubricant performance evaluation for metal forming applications with the following two most remarkable results:

(a) The construction of important digital characteristics (DC) of hot stamping process showing the evolutionary and complex features of tribological contact conditions via innovative 2D data mapping and 3D dynamic visualisation.

(b) The development of novel data-centric lubricant limit diagrams (LLD) – the holistic graphic description to intuitively and quantitatively reveal lubricant performance.

Metal forming is one of the most widely applied manufacturing techniques and the developed of innovative DC and LLD has provided new insights into the tribological contact and the lubrication behaviours during hot metal forming thus advancing scientific understand. Technologically, they are essential for the digital transformation of the metal forming industrial sector notably for saving energy via reducing friction, reducing the use of lubricant and the associated environmental impact, increasing performance and life of forming tools, selecting optimal lubricant for a specific process and providing recommendation on the development of new lubricants. Equally, it also provides information on the tribological interactions of different areas of a forming tool essential for developing selective surface treatment and coating for die surfaces.

The paper is clearly written with thoughtfully designed illustration and figures. The methodology is sound with sufficient details and the results are coherently presented and logically interpreted.

Hence, it is recommended for publication in the journal after minor correction:

a) English is clear but the use of the word 'respectively' in some places (Lines 113, 123, 194, 209) is not essential;

b) SV (Line 132) should be fully defined when it appears for the first time;

c) What do you mean by 'tribologically safe area' (Line 172)?

d) 'Performance grades' (Lines 182-3) should be defined when it appears for the first time or referred to Eq 1 under Methodology; and

e) Full definition should be given to all the parameters used in Eq 1.

Reviewer #2 (Remarks to the Author):

What are the noteworthy results?

The analysis of lubricant failure in FEM on a nodal level is a practical approach to improving a digitally supported process design. Especially for hot stamping of aluminium, lubrication failure due to adhesion is a relevant topic.

Such analysis can be performed in the post-processing stage of FEM, reducing the challenges of a coupled approach (contact conditions directly influence friction values during FEM calculation). Different lubricants, specially developed for aluminium hot stamping, show significant performance differences during friction testing.

While it is well known, that contact conditions vary greatly during forming processes, this work confirms their influence on the lubrication performance.

Will the work be of significance to the field and related fields? How does it compare to the established literature? If the work is not original, please provide relevant references.

The presented approach is overall promising and can be of great significance. While the prediction of adhesion is not relevant for cold sheet metal forming, it plays a major role in the hot stamping of aluminium. Predicting COFs and lubrication performance limits is a auspicious approach to improving the efficiency of the aluminium hot stamping process design.

However, the presented methods show weaknesses in the experimental friction testing and prediction model determination. The author has already published the underlying model for COF and failure prediction in an extended form [22], [23]

Does the work support the conclusions and claims, or is additional evidence needed?

Based on the DC and LLD, four lubricants show significant differences in their OPG. To validate these results, further experimental testing of the four lubricants in the hot stamping process is needed. Furthermore, additional friction tests are necessary to confirm the prediction model (for the explanation, please check "Is the methodology sound? Does the work meet the expected standards in your field?").

Are there any flaws in the data analysis, interpretation and conclusions? Do these prohibit publication or require revision?

-The Overall Performance Grade (OPG) is calculated as the average value of the performance grade (G). If this is done based on nodal values, an error due to local mesh size can be expected, as described in lines 202- 206. A calculation based on the surface area instead of nodal values should be introduced. This does not prohibit publication.

-The normalization of the sliding distance should be reconsidered. Since lubrication failure occurs after a relative sliding distance in mm, a normalization of the value makes it difficult to compare to other forming processes in the future. This does not prohibit publication.

Is the methodology sound? Does the work meet the expected standards in your field?

Minor comments:

-It is not explained how μ_d is determined during the calculation of performance grade (G).

Major comments:

The model used to calculate the COF is proposed in sources 22 and 23. Within the presented work, no information on the used model is given. After reading the relevant sources (18, 22, 23,...), the following questions arise, which lead to the conclusion, that the presented methodology does not meet the standard:

The model used to describe the lubricant failure is a step towards a "white-model" approach to the tribological system, aiming to describe the lubricant behaviour mathematically, based on physically driven assumptions. The mathematical model has a large number of parameters/constants that are derived out of experimental testing data (probably via MMSQE). The following points concerning the model are not clear:

1. Which experimental data is used to fit the model parameters/validate the model?

A separate set of experimental data should be used to determine the model parameters and validate the model. This is done within machine learning approaches by separating the data into learning/validation and testing data. The given information on the model (22, 23 and further work of the author) lead to the conclusion that the model was validated on test data, which was also part of the "training"/parameter determination process. Whether the model is valid to determine a COF for contact conditions that were not part of the experimental data is unclear.

2. Within lines 131-142 and sources [22] and [23], the importance of changing contact conditions and its influence on lubricant performance is highlighted. However, the presented data in "Extended Data Fig. 1" shows that only constant contact conditions were tested. This is in contrast to the presented work in [22] and [23].

The interface temperature data of the DC is the result of a thermomechanical simulation with multiple influencing factors. Within the prediction model/experimental testing, on the other hand, it is assumed that the temperature of the specimen holder corresponds directly with the interfacial temperature. Heat generation due to sliding and heat flux due to contacts are neglected. Since temperature plays a predominant role in lubricant failure a more accurate determination of contact temperatures during experiments (e.g. by FEM) is recommended. Example sources with the profound determination of contact temperatures are available: „Lubricant failure in sheet metal forming processes" Doctoral Thesis Emile van de Heide, 2002. , "Prediction of limits of lubrication in strip reduction testing", D.D. Olsson, N. Bay, J. L. Andreasen, 2004, "A study on the performance of environmentally benign lubricants at elevated temperatures in bulk metal forming", P. Groche 2015

Is there enough detail provided in the methods for the work to be reproduced?

The results cannot be reproduced with the given information.

- The forming process in which the data in Fig. 1 is presented is not explained. It is unclear whether the presented data are derived from multiple simulations or a single simulation. No information on the COF used in this/these simulations is given.
- No information on the four lubricants is given.

- Information on the parameter determination of the prediction model (presented in [22] and [23]) is missing.
- The friction testing conditions are not described.

Further Comments:

A radial bar chart to display this data makes it very hard to interpret this data. The chosen colours are too similar to distinguish. Effective strain and stress are not mentioned anymore and play no role in determining of lubricant performance within this work. Considering a large amount of space this diagram uses, the value for the author is marginal. The data is presented in Fig 2 a) in a much more straightforward way.

Line 115 – a range of 20 – 120 MPa is difficult to compare to the ranges given in Fig. 2a)

Line 132 – The short forms T, P, and SV are not introduced

Observing the maximum changes of tribological loads is important Fig. 2 c), however, is challenging to read and not necessary for the reader. The given table of max., med., and min. values are enough.

Line 179 – Add the short form for performance grade (G) to Fig. 3 so OPG and G can clearly be distinguished

Extended Data Fig. 1 a) –d) for the graphs showing tests at different temperatures, it is not clear which contact pressure and sliding speed are being used.

Extended Data Fig. 2 a) – adding gridlines enhances readability. Missing source! This has already been published in [23]

The paper title implies that the presented results apply to the entire field of metal forming. In sheet metal forming, lubricant failure due to adhesion is common for hot stamping of aluminium, not for cold sheet metal forming. Furthermore, the underlying model, presented in [22] and [23] is designed for a two-phase lubricant, specially developed for hot stamping of aluminium. It is not clear if the presented method is applicable for bulk metal forming. An adaptation of the paper title is needed to indicate the specific area of metal forming it is relevant for.

In order for this work to be published, the following aspects have to be improved:

- Details of the underlying prediction model including details on the parameter determination and model validation
- Improvement of the methodology for contact temperature determination of the friction test (specimen holder temperature \neq contact temperature)
- Friction testing with changing contact conditions (according to numbers presented in Fig. 2 c))

Reviewer #3 (Remarks to the Author):

- The novelty, significance, and benefits of this research are unclear to this reviewer.
- Many terms concerning 'data' were used without being defined clearly. What is the difference between 'data-centric', 'data-guided', and 'data-driven' approaches? What does 'digitally enhanced lubricant' mean?
- The introduction section of this paper is weak. It is unclear what specifically has motivated this work. The discussion on Industry 4.0 is fairly general. What is the knowledge gap filled by this work?
- The final discussion is lacking depth. No discussions are concerning the limitations and future work.

Comments from Reviewer #1:

Summary: This communication reports a novel approach for digitally enhanced lubricant performance evaluation for metal forming applications with the following two most remarkable results:

(a) The construction of important digital characteristics (DC) of hot stamping process showing the evolutionary and complex features of tribological contact conditions via innovative 2D data mapping and 3D dynamic visualisation.

(b) The development of novel data-centric lubricant limit diagrams (LLD) – the holistic graphic description to intuitively and quantitatively reveal lubricant performance.

Metal forming is one of the most widely applied manufacturing techniques and the developed of innovative DC and LLD has provided new insights into the tribological contact and the lubrication behaviours during hot metal forming thus advancing scientific understand. Technologically, they are essential for the digital transformation of the metal forming industrial sector notably for saving energy via reducing friction, reducing the use of lubricant and the associated environmental impact, increasing performance and life of forming tools, selecting optimal lubricant for a specific process and providing recommendation on the development of new lubricants. Equally, it also provides information on the tribological interactions of different areas of a forming tool essential for developing selective surface treatment and coating for die surfaces.

The paper is clearly written with thoughtfully designed illustration and figures. The methodology is sound with sufficient details and the results are coherently presented and logically interpreted. Hence, it is recommended for publication in the journal after minor correction:

Comment (a) English is clear but the use of the word ‘respectively’ in some places (Lines 113, 123,194, 209) is not essential;

Authors’ Response and modification to the manuscript: The authors appreciate this reviewer’s comments and the word ‘respectively’ in the mentioned places has been deleted as suggested.

Comment (b) SV (Line 132) should be fully defined when it appears for the first time;

Authors’ Response: The authors thank the reviewer for bringing this issue to our attention. Modification has been made as suggested.

Authors’ modification to the manuscript: On page 4, paragraph 4, we added:
‘contact condition, i.e., interfacial temperature (T), contact pressure (P) and sliding speed (SV), evolutions of each tool element’

Comment (c) What do you mean by ‘tribologically safe area’ (Line 172)?

Authors’ Response: ‘tribologically safe area’ means the lubricant has sufficient lubricity to protect the contact surfaces, which corresponds to the green area demonstrated in the LLD where the COF

value increase is less than 30% compared to the lubricated condition, leading to a satisfactory surface quality for ‘Class C surfaces’ in the automotive industry.

Authors’ modification to the manuscript: On page 6, paragraph 1, we added:

‘tribologically safe area, which means the lubricant has sufficient lubricity to protect the contact surfaces’

Comment (d) ‘Performance grades’ (Lines 182-3) should be defined when it appears for the first time or referred to Eq 1 under Methodology; and

Authors’ Response: The authors appreciate this reviewer’s comments and modification has been made as suggested.

Authors’ modification to the manuscript: On page 6, paragraph 2, we added:

‘performance grade (G) represents a quantified evaluation parameter of the lubricant performance under the applied contact conditions whilst considering the increase of COF value at the interface as a criteria. (see detailed calculation in Eq. (1), Methods)’

Comment (e) Full definition should be given to all the parameters used in Eq 1.

Authors’ Response: The authors appreciate this reviewer’s comments and full definitions of parameters in Eq 1 have been added as suggested.

Authors’ modification to the manuscript: On page 13, paragraph 1, we added:

‘ μ is the instantaneous coefficient of friction calculated by the interactive friction model, μ_l is the friction value under lubricated condition, μ_d is the friction value under dry sliding condition.’

Comments from Reviewer #2:

Summary: The analysis of lubricant failure in FEM on a nodal level is a practical approach to improving a digitally supported process design. Especially for hot stamping of aluminium, lubrication failure due to adhesion is a relevant topic. Such analysis can be performed in the post-processing stage of FEM, reducing the challenges of a coupled approach (contact conditions directly influence friction values during FEM calculation).

Different lubricants, specially developed for aluminium hot stamping, show significant performance differences during friction testing.

While it is well known, that contact conditions vary greatly during forming processes, this work confirms their influence on the lubrication performance.

Comment (a) The presented approach is overall promising and can be of great significance. While the prediction of adhesion is not relevant for cold sheet metal forming, it plays a major role in the hot stamping of aluminium. Predicting COFs and lubrication performance limits is an auspicious approach to improving the efficiency of the aluminium hot stamping process design. However, the presented methods show weaknesses in the experimental friction testing and prediction model determination. The author has already published the underlying model for COF and failure prediction in an extended form [22], [23].

Authors' Response: Many thanks for the acknowledgement of the significance of this work. In terms of the presented methods, the authors' response is shown below:

While the underlying model for COF and failure prediction has been presented and investigated in references [22] and [23], we have never employed the presented data guided approach in our previous work, which is the real novelty of this contribution. The present study emphasizes the digitally-enhanced evaluation of lubricant performance, and specifically investigates the lubricant failure based on big data analysis. The model from the prior studies was utilised and extended to a more complex and realistic test case whilst utilising the vast volumes of data extracted from the hot stamping process, and subsequent LLD demonstration and analysis.

Comment (b) Based on the DC and LLD, four lubricants show significant differences in their OPG. To validate these results, further experimental testing of the four lubricants in the hot stamping process is needed.

Authors' Response: The invaluable suggestion of this reviewer has been followed. The authors thank the reviewer for this comment, which has enabled to further demonstrate and highlight the impact of the work.

To validate the prediction results based on the DC and LLD, further experimental testing of the identified best lubricant (#3) was conducted in June 2022, by using an industry scale production system to perform hot stamping of a 'side beam'. The following tasks were conducted for this specific 'experimental testing' to validate the prediction of LLD.

1. Development of DC for the hot stamping of the 'side beam'. (Supplementary Fig. 4)
2. Identification of the most suitable lubricant candidate for this specific further experimental testing using LLD: lubricant #3 was identified for this specific experimental testing. (Supplementary Fig. 5)
3. Validation of the LLD prediction by experimental testing of the identified best lubricant (#3). (Supplementary Fig. 6)

Details of each task is shown below:

Task 1:

Tribologically-related data on the tool-workpiece interface were extracted from the experimentally verified FE simulations of the hot stamping of a side beam component. DC of this process was generated by plotting the probability of elements falling into specific ranges determined by contact pressure and normalised sliding distance, as shown in Supplementary Fig. 4.

Task 2:

The selection of the lubricant to be applied in the hot stamping of the 'side beam' was conducted using LLD. In addition, an environmentally friendly requirement was also raised by the end user. Therefore, lubricant #4 was not considered in this forming test as it is a graphite-based lubricant, which does not meet the environmental specifications. The most suitable lubricant candidate was selected from lubricants #1-3, which are water-based lubricants.

Based on DC and LLD analyses, the overall performance grade (OPG) of lubricants #1, #2 and #3 for this 'side beam' forming test were 78.4%, 75.8% and 97.3%. Data distribution of the lubricant failure region was also investigated for each lubricant, as shown in Supplementary Fig. 5. It can be observed that the dangerous region for the target forming of the side beam was the blank holder and side wall areas. Therefore, lubricant #3 was the best candidate to be applied during the hot stamping of the side beam, with OPG value up to 97.3%. Possible failure of the lubricant was found in the blank holder and side wall areas only.

Task 3:

Supplementary Fig. 6 demonstrates comparisons of surface quality of the formed side beam component between unlubricated condition and lubricated condition with the application of lubricant #3 on the tooling surface. The forming test equipment (side view and front view of the forming tool with integrated heating device) is shown below. Under unlubricated condition, severe scratches were found on the side wall / blank holding area (Supplementary Fig. 6a). After the application of the lubricant #3, the surface quality was significantly improved (Supplementary Fig. 6b). Surface scratches were negligible, which is consistent with the LLD predictions.

[REDACTED]

Authors' modification to the manuscript: On page 10, paragraph 2, we added:

'To validate the evaluation results based on the DC and LLD, further experimental testing of the identified most suitable lubricant was conducted by using an industry scale production system to perform hot stamping of a 'side beam' component (Supplementary Fig. 4). An environmentally friendly requirement was raised by the end user. Thus, lubricant #4 was not considered in this forming test as it is a graphite-based lubricant, which does not meet the environmental specifications. The most suitable lubricant candidate was selected from lubricants #1-3, which are water-based lubricants. Based on DC and LLD analyses, the OPG of lubricants #1, #2 and #3 for this 'side beam' forming test were 78.4%, 75.8% and 97.3%. Data distribution of the lubricant failure region was also investigated for each lubricant (Supplementary Fig. 5). Therefore, lubricant #3 was the best candidate to be applied during this forming process. Under unlubricated condition, severe scratches were found on the side wall / blank holding area (Supplementary Fig. 6a). After the application of the lubricant #3, the surface quality was significantly improved (Supplementary Fig. 6b). Surface scratches were negligible, which is consistent with the LLD predictions.'

Supplementary Fig. 4. Further experimental validation of the LLD predictions by performing hot stamping of a ‘side beam’ component. a) Geometry of the side beam and DC of the hot stamping process of the side beam, b) Comparisons between DC of the side beam and overall hot stamped components. Forming of the ‘side beam’ experiences relatively short normalised sliding distances of generally less than 0.3.

Supplementary Fig. 5. Identification of the most suitable lubricant candidate for this experimental validation and data distribution analysis of the lubricant failure region in the forming of the side beam. a) lubricant #1, b) lubricant #2 and c) lubricant #3. The dangerous region for the forming test of the ‘side beam’ was the blank holder and side wall areas.

[REDACTED]

Comment (c) The Overall Performance Grade (OPG) is calculated as the average value of the performance grade (G). If this is done based on nodal values, an error due to local mesh size can be expected, as described in lines 202-206. A calculation based on the surface area instead of nodal values should be introduced. This does not prohibit publication.

Authors' Response: The authors appreciate this comment and our response to the comment enables us to clarify how values are extracted and to show that we have taken all precautions of the case to minimise the sources of error:

1. The calculation of the performance grade (G) involves contact condition data extracted from FE simulations, e.g., interfacial temperature and relative sliding speed data, derived from a node basis, and contact pressure, from an elemental average basis.
2. A mesh size sensitivity study was conducted for each simulation. The studied mesh size ranged from the blank thickness value (usually 1.5 or 2 mm) to around 10 mm (shell elements), and finally a mesh size of the thickness value or 30% larger than the thickness was selected to ensure convergence and accuracy of simulation results. All simulations were experimentally-verified, showing that close agreements (with less than 5% errors) have been

achieved between the experimental results and simulated predictions, which further ensure the robustness of the extracted data and calculation of OPG.

Comment (d) The normalization of the sliding distance should be reconsidered. Since lubrication failure occurs after a relative sliding distance in mm, a normalization of the value makes it difficult to compare to other forming processes in the future. This does not prohibit publication.

Authors' Response: The authors thank the reviewer for this comment and provide a full response and some further clarification below:

In this study, the normalised sliding distance is determined by the relative sliding distance divided by the forming stroke of the corresponding forming process, which is used to make processes with different forming strokes comparable. But it should be noted that the lubricant performance is evaluated and the COF evolution is calculated based on the relative sliding distance in 'mm'.

The normalization of sliding distance was used for the following reasons:

1. Data availability. The metadata set of hot stamping processes used in the present study was provided by a cloud-based repository for knowledge transfer and data sharing. The absolute values of some processing parameters are protected upon the request of data contributors.
2. Data compatibility. The metadata set includes multiple FE simulations with various forming strokes ranging from 15 mm to 200 mm. The normalisation of relative sliding distance enables data collected from different simulations to be compatible and comparable.

Authors modification to the manuscript: On page 4, paragraph 1, we added:

'As each forming process of the component has a unique forming stroke, the normalised sliding distance is introduced as the ratio of the absolute relative sliding distance and the forming stroke of the corresponding forming process, to allow processes with different forming strokes comparable.'

Comment (e) It is not explained how μ_d is determined during the calculation of performance grade (G).

Authors' Response: The authors appreciate this comment and modification has been made as suggested to explain how μ_d and μ_l are determined in the LLD calculation.

Authors' modification to the manuscript: On page 13, paragraph 1-2, we added:

' μ_d and μ_l are determined based on friction testing results and modelled by using the Arrhenius equation to represent the temperature effects, as described by the following two equations:

$$\mu_d = \mu_{d0} \exp\left(-\frac{Q_d}{RT}\right), \mu_l = \mu_{l0} \exp\left(-\frac{Q_l}{RT}\right)$$

where μ_{d0} and μ_{l0} are model constants, Q_d and Q_l are activation energy for dry sliding and lubricated conditions, respectively. These parameters were calibrated against the friction testing results under different temperature conditions.'

Comment (f) Within lines 131-142 and sources [22] and [23], the importance of changing contact conditions and its influence on lubricant performance is highlighted. However, the presented data in

“Extended Data Fig. 1” shows that only constant contact conditions were tested. This is in contrast to the presented work in [22] and [23].

Authors’ Response: The authors thank the reviewer for this helpful comment, which has been fully clarified below:

According to the conclusions of reference [22] and [23], the calibration of the interactive friction model against testing results under constant contact conditions enables an accurate prediction of COF under complex loading conditions. This advantage of the calibrated model, with time-dependent equations, enables the underlying lubrication mechanism transformation leading to lubricant breakdown to be determined through simplified constant contact conditions.

Further information and the corresponding model response under complex loading conditions for the two-phase lubricant are shown in Supplementary Fig. 2. In the load increase test, there was an abrupt change of contact load from 5 N to 10 N at the sliding distance of 20 mm. Due to this abrupt change, the thickness changing rate of both liquid lubricant film and solid tribo-layer also experienced a sudden drop accordingly, as shown in Supplementary Fig. 2a, leading to deviations of film thickness away from the constant loading curve and, thus, earlier breakdown was observed in the load increase test as was successfully captured and predicted by the calibrated interactive friction model. Similar phenomena could also be found under the changes of speed and temperature. According to this advanced and inherent feature of the friction model, the COF evolution could be determined under complex contact conditions, with good agreements, even though the model parameters were calibrated against testing results under constant contact conditions.

As this feature has been thoroughly investigated and the corresponding interactive friction models have been established in [22] and [23], the same model and testing procedure for lubricants #1-4 was not included in this work to avoid repetition. In addition, an industrial scale forming test of a ‘side beam’ were performed to validate the friction models established in this study. (Details are shown in response to Comment (b))

Authors’ modification to the manuscript: On page 12, paragraph 1, we added:

‘According to the conclusions of the references^{35,36}, the mechanism-based and time-dependent interactive friction model can accurately represent and predict lubricant behaviour under complex loading conditions after the model parameters are calibrated against testing results under constant contact conditions. This advantage of the calibrated model, with time-dependent equations, enables the underlying lubrication mechanism transformation leading to lubricant breakdown to be determined through simplified constant contact conditions. Further information and the corresponding model response under complex loading conditions for the two-phase lubricant are presented in Supplementary Fig. 2 to explain this advanced feature of the interactive friction model.’

Supplementary Fig. 2. Model responses in evolutions of film thickness and thickness changing rate for the two-phase lubricant under complex loading conditions. a) load increase test; b) speed increase test. In the load increase test, there was an abrupt change of contact load from 5 N to 10 N at the sliding distance of 20 mm. Due to this abrupt change, the thickness changing rate of both liquid lubricant film and solid tribo-layer also experienced a sudden drop accordingly, leading to deviations of film thickness away from the constant loading curve and, thus, earlier breakdown was observed in the load increase test as was successfully captured and predicted by the calibrated interactive friction model. Similar phenomena could also be found under the changes of speed and temperature. According to this advanced and inherent feature of the friction model, the COF evolution could be determined under complex contact conditions, with good agreements, even though the model parameters were calibrated against testing results under constant contact conditions.

Comment (g) Which experimental data is used to fit the model parameters/validate the model? A separate set of experimental data should be used to determine the model parameters and validate the model. This is done within machine learning approaches by separating the data into learning/validation and testing data. The given information on the model (22, 23 and further work of the author) lead to the conclusion that the model was validated on test data, which was also part of the “training”/parameter determination process. Whether the model is valid to determine a COF for contact conditions that were not part of the experimental data is unclear.

Authors’ Response: The authors appreciate this comment from reviewer and the full clarification is provided below:

Different sets of experimental data were used to determine the model parameters and validate the model in the work of [22] and [23]. Namely, the testing data obtained under constant loading conditions were used to determine the model parameters. A separate set of experimental data obtained under complex loading conditions were used to validate the model. Close agreement has been achieved between experimental results and model predictions under complex loading conditions. Therefore, it is validated that the established model (after model parameter determination) can accurately predict the COF evolution under complex contact conditions that is not part of the experimental data in the constant friction tests.

According to this conclusion drawn from the work of [22] and [23], friction testing results of four lubricant candidates under constant loading conditions were utilised to calibrate the interactive model parameters in this study. This has been clarified in the revised manuscript.

Authors’ modification to the manuscript: On page 11, paragraph 4, we added:

‘Friction testing results of four lubricant candidates under constant loading conditions were utilised to calibrate the interactive model parameters and close agreements have been achieved between modelling and experimental results’

Comment (h) The interface temperature data of the DC is the result of a thermomechanical simulation with multiple influencing factors. Within the prediction model/experimental testing, on the other hand, it is assumed that the temperature of the specimen holder corresponds directly with the interfacial temperature. Heat generation due to sliding and heat flux due to contacts are neglected. Since temperature plays a predominant role in lubricant failure a more accurate determination of contact temperatures during experiments (e.g. by FEM) is recommended. Example sources with the profound determination of contact temperatures are available: “Lubricant failure in sheet metal forming processes” Doctoral Thesis Emile van de Heide, 2002, “Prediction of limits of lubrication in strip reduction testing”, D.D. Olsson, N. Bay, J. L. Andreasen, 2004, “A study on the performance of environmentally benign lubricants at elevated temperatures in bulk metal forming”, P. Groche 2015.

Authors' Response: The authors appreciate this comment and have investigated the temperature at the contact interface by using FE simulations (i.e., Abaqus) as suggested by the reviewer:

According to the results (Supplementary Fig. 7 and 8), the temperature deviation between the contact interface and the measure point is less than 1.3% of the nominated value, which is acceptable and presents minor effects on the final lubricant evaluation results. This can be expected due to the thin aluminium blank specimen (1.6 mm) and the small contact area between the pin and blank ($\sim 0.78 \text{ mm}^2$), which would generate limited frictional heat during sliding. Details of FE analysis are shown as follows.

Frictional heat generation due to relative sliding, conductance between the hot aluminium blank and cold steel pin and convection to the ambient air were considered in this temperature distribution analysis. Radiation was negligible due to the relatively low emissivity coefficient of aluminium. The interfacial heat transfer coefficient (IHTC) between AA7075 aluminium alloy and tool steel has been investigated by Liu et al.¹, which indicates a stable value of IHTC as the contact pressure is greater than 10 MPa. The point of temperature measurement by thermo-couples embedded in the specimen holder was located at the central plane across the blank thickness. A constant temperature boundary condition is, therefore, assumed at the central plane of the blank due to the insulation of thermal box and short sliding time.

Supplementary Fig. 7 shows the simulation results of interfacial temperature distribution during lubricated sliding. The average temperature is calculated based on this distribution with the error bar (enveloped area) indicating the standard deviation, as shown in Supplementary Fig. 8.

Reference:

1. Liu, X. *et al.* Determination of the interfacial heat transfer coefficient for a hot aluminium stamping process. *J. Mater. Process. Technol.* **247**, 158–170 (2017).

Authors' modification to the manuscript: We added Supplementary Fig. 7 and Fig. 8 to show that the temperature deviation between the contact interface and the measure point is negligible and presents minor effects on the final lubricant evaluation results.

Supplementary Fig. 7. Interfacial temperature distribution across the sliding wear track investigated by using Abaqus. Under the initial blank temperature of 350°C, relative sliding speed of 30 mm/s and contact pressure of 19 MPa, as a function of the sliding time: (a) 0s, (b) 0.05s, (c) 0.1s and (d) 0.15s. Frictional heat generation due to relative sliding, conductance between the hot aluminium blank and cold steel pin and convection to the ambient air were considered in this temperature distribution analysis. Radiation was negligible due to the relatively low emissivity coefficient of aluminium. The interfacial heat transfer coefficient (IHTC) between AA7075 aluminium alloy and tool steel indicates a stable value of IHTC as the contact pressure is greater than 10 MPa. A constant temperature boundary condition is assumed at the central plane of the blank, where thermos-couples were located, due to the insulation of thermal box and short sliding time.

Supplementary Fig. 8. Simulation results of the actual interfacial temperature under different testing conditions. Scatter represents the average value; envelope represents the standard deviation. The temperature deviation between the contact interface and the measure point is less than 1.3% of the nominated value, which is acceptable and presents minor effects on the final lubricant evaluation results. This can be expected due to the thin aluminium blank specimen (1.6 mm) and the small contact area between the pin and blank ($\sim 0.78 \text{ mm}^2$), which would generate limited frictional heat during sliding.

Comment (i) The forming process in which the data in Fig. 1 is presented is not explained. A radial bar chart to display this data makes it very hard to interpret this data. The chosen colours are too similar to distinguish. Effective strain and stress are not mentioned anymore and play no role in determining of lubricant performance within this work. Considering a large amount of space this diagram uses, the value for the author is marginal. The data is presented in Fig 2 a) in a much more straightforward way.

Authors' Response: The authors appreciate this comment and has followed the suggestion from the reviewer.

Authors' modification to the manuscript: Fig. 1 has been deleted.

Comment (j) It is unclear whether the presented data are derived from multiple simulations or a single simulation. No information on the COF used in this/these simulations is given.

Authors' Response: The authors appreciate this comment. The presented data are indeed derived from multiple simulations. All the simulations are experimentally verified to ensure the robustness of the extracted FE simulation results and veracity of data. Although constant friction value was used in the individual simulation, the vast volume of the metadata set which includes more than 15 billion data enables the sensible observation and analysis of characteristics regarding tribologically-related investigations. To clarify this, modification has been made as suggested.

Authors' modification to the manuscript: On page 3, paragraph 3, we added:

'The DC of the hot stamping process is developed based on the metadata, which is provided by a cloud-based repository for knowledge transfer and data sharing, which contains open access data sets for research purposes²⁴⁻²⁶. This metadata set containing multiple simulations has been contributed by developers whose work has been previously reported in peer-reviewed studies over the past 15 years^{17,27-30}. The entire metadata set covers the manufacturing processes of a wide range of hot-stamped components from the FE analysis spanning over 20 types of evolutionary thermo-mechanical parameters including interfacial temperature, contact pressure, sliding speed, effective stress and strain, etc. All of the involved simulations have been experimentally verified to ensure the robustness of the extracted thermo-mechanical parameters and veracity of data.'

Comment (k) No information on the four lubricants is given.

Authors' Response: The authors appreciate this comment. Key parameters and information on the four lubricants is provided in Supplementary Table 1. The commercial name and exact composition of the lubricants investigated here are not essential for the demonstration of the method and we prefer to focus on the known important physical properties, which are provided here.

Authors' modification to the manuscript: We added Supplementary Table 1 to demonstrate the important physical properties of the four lubricants.

Comment (l) Information on the parameter determination of the prediction model (presented in [22] and [23]) is missing.

Authors' Response: The authors appreciate this comment and calibration of parameters were conducted by the in-situ friction modelling program embedded in Tribo-Mate testing system (referred to [22] and [23]). The calibration process is an optimisation of an objective function, $f(x)$, which is determined by the deviations between the predicted and experimental results, as expressed by the following equation:

$$f(x) = \sum_{i=1}^m w_i (\mu_i^p - \mu_i^e)^2$$

where $f(x)$ is the sum of residuals for COF values, x ($x = [x_1, x_2, \dots, x_s]$) represents the model parameters and s is the number of the model parameters to be calibrated. μ_i^p and μ_i^e are the predicted and experimental results for the COF at the same sliding distances. w is a weighting function and m is the total number of data points considered. The optimised parameters of the established interactive friction models for four lubricants are displayed in Supplementary Tables 2-5. Modifications have been made to the revised manuscript as suggested.

Authors' modification to the manuscript: On page 12, paragraph 2, we added:

'Calibration of model parameters were conducted by the in-situ friction modelling program embedded in Tribo-Mate testing system^{35,36}. The optimised parameters of the established interactive friction models for four lubricants are displayed in Supplementary Tables 2-5.'

We added Supplementary Table 2-5 to demonstrate the calibrated/optimised model parameters for the four lubricants.

Comment (m) The friction testing conditions are not described.

Authors' Response and modification to the manuscript: The authors appreciate this comment and detailed testing conditions have been displayed in the table, below the graphs showing the friction testing results, for each lubricant in Supplementary Fig. 1(a-d).

Comment (n) Line 115 – a range of 20 – 120 MPa is difficult to compare to the ranges given in Fig. 2a).

Authors' Response: Modification has been made to the manuscript to link the text directly to the information provided by Fig. 2a as suggested.

Authors' modification to the manuscript: On page 4, paragraph 2, we modified the description:

'Approximately 12.2% of elements fall into the range of 20-140 MPa with the normalised sliding distance lower than 0.1.'

Comment (o) Line 132 – The short forms T, P, and SV are not introduced.

Authors' Response: The authors thank the reviewer for bringing this issue to our attention. Modification has been made as suggested.

Authors' modification to the manuscript: On page 4, paragraph 4, we added:
'contact condition, i.e., interfacial temperature (T), contact pressure (P) and sliding speed (SV), evolutions of each tool element'.

Comment (p) Observing the maximum changes of tribological loads is important Fig. 2 c), however, is challenging to read and not necessary for the reader. The given table of max., med., and min. values are enough.

Authors' Response and modification to the manuscript: Fig. 2c has been deleted as suggested and only table of max., med., and min. values is shown in the revised manuscript.

Comment (q) Line 179 – Add the short form for performance grade (G) to Fig. 3 so OPG and G can clearly be distinguished.

Authors' Response and modification to the manuscript: The authors appreciate this comment and the reviewer's suggestion has been followed in the revised manuscript.

Comment (r) Extended Data Fig. 1 a) –d) for the graphs showing tests at different temperatures, it is not clear which contact pressure and sliding speed are being used.

Authors' Response and modification to the manuscript: Detailed testing conditions have been displayed in the table, below the graphs showing the friction testing results, for each lubricant in Supplementary Fig. 1(a-d).

Comment (s) Extended Data Fig. 2 a) – adding gridlines enhances readability. Missing source! This has already been published in [23].

Authors' Response: The authors thank the reviewer for pointing out this issue. To avoid confusion and similar appearance with reference [23], the figure has been replaced by a new one, as shown in Supplementary Fig. 3. In addition, a radial bar chart of the hot stamping metadata set is added to help readers understand that this example element evolution is extracted from the hot stamping process.

Authors' modification to the manuscript: We modified Supplementary Fig. 3.

Comment (t) The paper title implies that the presented results apply to the entire field of metal forming. In sheet metal forming, lubricant failure due to adhesion is common for hot stamping of aluminium, not for cold sheet metal forming. Furthermore, the underlying model, presented in [22] and [23] is designed for a two-phase lubricant, specially developed for hot stamping of aluminium. It is not clear if the presented method is applicable for bulk metal forming. An adaptation of the paper title is needed to indicate the specific area of metal forming it is relevant for.

Authors' Response: While the authors believe that the presented methodology is indeed general and can be extended to other forming processes, we somewhat agree that the title may be misleading as this has only been demonstrated for hot stamping.

Authors' modification to the manuscript: We modified the title of the paper to become:

'Digitally-enhanced lubricant evaluation scheme for hot stamping applications'

Comments from Reviewer #3:

Comment (a) The novelty, significance, and benefits of this research are unclear to this reviewer.

Authors' Response: The authors appreciate this comment and the clarification is shown below:

Digitally-enhanced technologies are transforming every aspect of manufacturing, including the metal forming industry. Networks of sensors, digital twins and newly emerging Cloud FEA technologies yield data at unprecedented scales, providing comprehensive information on complex processes that had previously been impossible to capture, analyse and utilise in practice. The present work elaborates a data-centric method for lubricant evaluation and development in metal forming applications, with the hot stamping process as a case study.

Forming processes are one of the most widely used manufacturing crafts and aluminium is a very commonly utilized material for such processes. Appropriate lubricant selection is essential to ensure products are produced with adequate surface quality whilst extending tooling life, with failure to accurately address these issues leading to production losses and product re-work. Lubricant failure is widely observed due to the severe and highly transient interfacial conditions in metal forming, with few options being available to industry other than trial and error and simplified FE simulations which are unable to adapt to constantly changing parameters and production volume effects. To comprehensively characterize this lubricant phenomenon and make the most sensible selection of lubricant applied in the target process, the simulation of lubricant changes throughout the entire lifecycle is required, and the proposed method to achieve this is the introduction of a novel data-centric lubricant evaluation method driven by mechanism-based accurate theoretical models. The proposed digitally-enhanced approach is adaptable to other metal forming techniques. Finally, the use of such methods has seen significant interest by the lubricant industry as it seeks to adapt their products to better address their industrial customer demand whilst minimising cost. By determining the lubricant performance across vast test cases which are only realistically possible through a mathematical and data-centric approach, product optimisation is possible whilst minimising costly and time-consuming experimental testing across all conditions.

Comment (b) Many terms concerning 'data' were used without being defined clearly. What is the difference between 'data-centric', 'data-guided', and 'data-driven' approaches? What does 'digitally enhanced lubricant' mean?

Authors' Response: The authors appreciate this comment. The usage of 'data-centric', 'data-guided' and 'data-driven', which have similar meanings, may confuse the readers. Thus, modification has been made as suggested and only 'data-centric' is used in the revised manuscript.

Authors' modification to the manuscript: On page 4, paragraph 4, we added:

'Here 'data-centric' indicates a significant transformation from traditional 'experience-oriented' approach for how research is conducted and information is processed. It encompasses extensive data collection and state-of-art data storage techniques to extract insightful data analytics and

comprehensively accelerate the implementation of novel ideas into applications. This provides an unprecedented opportunity in complementing traditional experience-oriented research approaches conventionally used in fields such as manufacturing with new concepts and emerging methods deduced from data.'

We replaced 'data-guided' and 'data-driven' with 'data-centric' in the revised manuscript.

Authors' Response: 'Digitally enhanced lubricant evaluation' means applying this data-centric approach in the performance evaluation of lubricants in various fields and applications.

Authors' modification to the manuscript: On page 5, paragraph 2, we added:

'The performance of four lubricants developed for the hot aluminium stamping process were evaluated following the data-centric approach and comprehensively demonstrated by LLDs, which is the digitally-enhanced lubricant evaluation process.'

Comment (c) The introduction section of this paper is weak. It is unclear what specifically has motivated this work. The discussion on Industry 4.0 is fairly general. What is the knowledge gap filled by this work?

Authors' Response and modification to the manuscript: The authors thank the reviewer for bringing this issue to our attention and modifications have been made to enhance the 'Introduction' section (on pages 1-3) as suggested.

Comment (d) The final discussion is lacking depth. No discussions are concerning the limitations and future work

Authors' Response and modification to the manuscript: The authors appreciate this comment and modifications have been made in the final 'Discussion' section (on pages 10-11) as suggested.

REVIEWER COMMENTS

Reviewer #2 (Remarks to the Author):

What are the noteworthy results?

The analysis of lubricant failure in FEM on a nodal level is a practical approach to improve a digitally supported process design. Especially for hot stamping of aluminium, lubrication failure due to adhesion is a relevant topic.

Such analysis can be performed in the post-processing stage of FEM, reducing the challenges of a coupled approach (contact conditions directly influence friction values during FEM calculation). Different lubricants, specially developed for hot stamping of aluminium, show large performance differences during friction testing.

While it is well known, that contact conditions vary greatly during forming processes, this work confirms their influence on the lubrication performance.

Will the work be of significance to the field and related fields? How does it compare to the established literature? If the work is not original, please provide relevant references.

The presented approach is overall very promising and can be of great significance. While the prediction of adhesion is not relevant for cold sheet metal forming, it plays a major role in hot stamping of aluminium. The prediction of COFs and lubrication performance limits is a promising approach to improve the efficiency of the aluminium hot stamping process design.

Does the work support the conclusions and claims, or is additional evidence needed?

The experimental work added (Supp. Fig. 4 – 6) increases the quality of the presented work. The purpose of the additional experimental work is to prove, that the presented method of OPG determination is a viable way to grade the suitability of lubricants for a given hot stamping process. The additional testing is missing an important part though. In the given data only a comparison between “no lubricant” and “Lubricant 3” is given. The increase in performance when using “lubricant 3” instead of “no lubricant” gives no information on if the method of OPG is viable. The test does not confirm, that lubricant 3 is more suitable than lubricant 1. It only shows that lubricant three is suitable. It is possible that OPG does not work and all three lubricants are viable. The tests need to be performed with all three lubricants, in order to see if the differences in OPG can be confirmed experimentally. It is necessary to validate that the method of OPG can identify lubricants, which are suitable and lubricants, which are NOT suitable for the forming process.

Are there any flaws in the data analysis, interpretation and conclusions? Do these prohibit publication or require revision?

-The Overall Performance Grade (OPG) is calculated as the average value of the performance grade (G). If this is done based on nodal values, an error due to local mesh size can be expected, as described in lines 202- 206. A calculation based on the surface area, instead of nodal values should be introduced instead. This does not prohibit publication.

This has been noted in the first review already the authors commented on it and made a sensitivity study and thorough definition of meshsize does not address this issue. Due to deformation, the meshsize can locally differ greatly from its original, even special

distribution. Additionally some FE-Software automatically adapt the meshsize locally. This issue needs to be addressed in the future work.

-The normalization of the sliding distance should be reconsidered. Since failure of lubricant takes place after a relative sliding distance in mm, a normalization of the value makes it difficult to compare to other forming processes in the future. This does not prohibit publication.

Please include an explanation in the manuscript regarding the fact that: The calculation of PG is based on sliding distance in mm, not based on normalized sliding length.

Is the methodology sound? Does the work meet the expected standards in your field?

Yes

Is there enough detail provided in the methods for the work to be reproduced?

With additional information found in past publications the tests are reproducible.

Further Comments:

In order for this work to be published, the following aspects have to be improved:

- Extend the experimental work in (Supp. Fig. 4 – 6) by the investigation of all lubricants.**
- Include information regarding the usage of sliding distance in mm for calculation of PG**

Reviewer #3 (Remarks to the Author):

The authors have addressed this reviewer's concerns. Overall, it is an interesting work.

Comments from Reviewer #2:

Comment (a) The experimental work added (Supp. Fig. 4 – 6) increases the quality of the presented work. The purpose of the additional experimental work is to prove, that the presented method of OPG determination is a viable way to grade the suitability of lubricants for a given hot stamping process. The additional testing is missing an important part though. In the given data only a comparison between “no lubricant” and “Lubricant 3” is given. The increase in performance when using “lubricant 3” instead of “no lubricant” gives no information on if the method of OPG is viable. The test does not confirm, that lubricant 3 is more suitable than lubricant 1. It only shows that lubricant three is suitable. It is possible that OPG does not work and all three lubricants are viable. The tests need to be performed with all three lubricants, in order to see if the differences in OPG can be confirmed experimentally. It is necessary to validate that the method of OPG can identify lubricants, which are suitable and lubricants, which are NOT suitable for the forming process.

Authors’ response: The invaluable suggestion of this reviewer has been followed. The authors thank the reviewer for this comment, and further experimental testing of all three lubricants were conducted to validate the presented method of OPG and LLD.

Supplementary Fig. 6 (see below) demonstrates comparisons of surface quality of the formed ‘side beam’ component between lubricated conditions with application of different lubricants #1-3. Two representative positions on the surface of the formed component are selected and zoomed in figures are demonstrated. When lubricant #1 or #2 was applied, moderate scratches were observed on the side wall/blank holding area (Supplementary Fig. 6a and 6b). While after the application of lubricant #3, scratches were negligible (Supplementary Fig. 6c) and surface quality was improved and desirable compared to #1 and #2, which is consistent with the LLD predictions.

Authors’ modification to the manuscript: On page 10, paragraph 2, we added:

‘When lubricant #1 or #2 was applied, moderate scratches were observed on the side wall/blank holding area (Supplementary Fig. 6a and 6b). While after the application of lubricant #3, scratches were negligible (Supplementary Fig. 6c) and surface quality was improved and desirable compared to #1 and #2, which is consistent with the LLD predictions.’

Comment (b) The Overall Performance Grade (OPG) is calculated as the average value of the performance grade (G). If this is done based on nodal values, an error due to local mesh size can be expected, as described in lines 202- 206. A calculation based on the surface area, instead of nodal values should be introduced instead. This does not prohibit publication.

This has been noted in the first review already the authors commented on it and made a sensitivity study and thorough definition of meshsize does not address this issue. Due to deformation, the meshsize can locally differ greatly from its original, even special distribution.

Additionally some FE-Software automatically adapt the meshsize locally. This issue needs to be addressed in the future work.

Authors' response: The authors thank the reviewer for bringing this point to our attention. The effects of localised meshsize will be further investigated in the future work.

Comment (c) The normalization of the sliding distance should be reconsidered. Since failure of lubricant takes place after a relative sliding distance in mm, a normalization of the value makes it difficult to compare to other forming processes in the future. This does not prohibit publication.
Please include an explanation in the manuscript regarding the fact that: the calculation of PG is based on sliding distance in mm, not based on normalized sliding length.

Authors' response and modification to the manuscript: The authors appreciate this comment and on page 6, paragraph 2, we added:

'It should be noted that both COF values and the performance grade are calculated based on the relative sliding distance (in 'mm') instead of the normalised sliding distance.'

[REDACTED]